# The impact of loneliness on depression, mental health, and physical well-being

**Oluwasegun Akinyemi**[1]*, **Waliah Abdulrazaq**[2], **Mojisola Fasokun**[3],
**Fadeke Ogunyankin**[4], **Seun Ikugbayigbe**[5], **Uzoamaka Nwosu**[2], **Miriam Michael**[6],
**Kakra Hughes**[7], **Temitope Ogundare**[8]

1 The Clive O Callender Outcomes Research Center, Howard University College of Medicine, Washington, District of Columbia, United States of America, 2 Department of Psychiatry and Behavioral Sciences, Howard University College of Medicine, Washington, District of Columbia, USA, 3 Department of Epidemiology, University of Alabama at Birmingham, Birmingham, Alabama, United States of America, 4 Department of Research Data Science and Analytics, Cook Children's Health Care System: Cook Children's Medical Center, Fort Worth, Texas, United States of America, 5 Department of Biological Sciences, Eastern Illinois University, Charleston, Illinois, United States of America, 6 Department of Internal Medicine, Howard University College of Medicine, Washington, District of Columbia, United States of America, 7 Department of Surgery, Howard University College of Medicine, Washington, District of Columbia, United States of America, 8 Department of Psychiatry, Boston University School of Medicine, Boston, Massachusetts, United States of America

* austineakinyemi@gmail.com

## Abstract

### Background

Loneliness is a growing public health concern, with substantial implications for mental and physical health. Despite increasing attention, it remains an underrecognized determinant of health outcomes in population-based research.

### Objective

This study evaluates the association between loneliness and key health outcomes, including depression diagnosis, poor mental health days, and poor physical health days, using a nationally representative sample.

### Methods

We analyzed Behavioral Risk Factor Surveillance System (BRFSS) data from 2016 to 2023. Loneliness was measured with the question, "How often do you feel lonely?" and categorized into five levels: Always, Usually, Sometimes, Rarely, and Never. We estimated average treatment effects (ATE) using inverse probability weighting (IPW), adjusting for sociodemographic characteristics and incorporating BRFSS sampling weights and fixed effects for state, and year.

**Data availability statement:** All relevant data are within the paper and its Supporting information files.

**Funding:** This project was supported (in part) by the National Institute on Minority Health and Health Disparities of the National Institutes of Health under Award Number 2U54MD007597. The content is solely the responsibility of the authors and does not necessarily represent the official views of the National Institutes of Health.

**Competing interests:** The authors have declared that no competing interests exist.

## Results

The study included 47,318 individuals, predominantly White (73.3%), female (62.1%), and aged 18–64 years (72.1%). Over 80% of participants reported some degree of loneliness. Compared to those who reported "Never" being lonely, individuals who reported being "Always" lonely had a significantly higher predicted probability of depression (50.2% vs. 9.7%, ATE = +40.5 percentage points, p < 0.001), 10.9 more poor mental health days, and 5.0 more poor physical health days per month (all p < 0.001). Disparities were evident across sex, race/ethnicity, and age. Women consistently reported more poor mental health days than men across most loneliness levels. Black individuals reporting loneliness had significantly lower probabilities of depression and fewer poor mental health days than White peers. Older adults (>64) experienced more poor physical health days than younger adults across all loneliness categories.

## Conclusion

Loneliness is a strong and independent predictor of depression and poor health outcomes. Public health interventions aimed at addressing loneliness—especially among high-risk subgroups—are critical to improving mental and physical well-being at the population level.

## Introduction

Loneliness has emerged as a significant public health concern in the United States, with profound implications for mental and physical health [1–3]. Defined as the subjective feeling of being socially disconnected or lacking meaningful interpersonal relationships [1], loneliness affects approximately 36% of Americans, rising to 61% among young adults and 51% among mothers with young children [4]. Women report loneliness at higher rates than men, and the phenomenon cuts across demographic and socioeconomic lines [5,6]. Critically, loneliness has been linked to a 26% increase in premature mortality, a risk comparable to smoking and obesity, underscoring its urgency as a public health priority [1,4].

The health consequences of loneliness are wide-ranging and well-documented [1,7]. Previous studies have established strong associations between loneliness and depression, identifying a bidirectional relationship where loneliness both contributes to and is exacerbated by depressive symptoms [8–10]. Lonely individuals are estimated to have a 15–30% higher risk of developing depressive disorders [11,12]. On average, lonely individuals report two to three more poor mental health days per month [10,13]. Physical health is similarly affected; loneliness contributes to one to two additional poor physical health days monthly, often linked to inflammation, immune dysfunction, and heightened risk of chronic diseases such as hypertension and cardiovascular disorders [10,14,15].

Despite growing recognition of loneliness as a serious health threat, previous studies have limitations that restrict our understanding of its full impact [3]. Much of the existing literature has relied on cross-sectional or correlational designs, limiting causal inference [3,16]. In addition, many studies use localized or non-representative samples, making it difficult to generalize findings across diverse populations in the U.S [16,18]. Moreover, the complex interplay between loneliness and its health outcomes, particularly across different demographic subgroups, has not been adequately explored [17,18]. Key social determinants of health such as race/ethnicity, income, education, and access to care are often overlooked, leaving gaps in our understanding of how loneliness disproportionately affects certain populations [3,16].

This study addresses these limitations by leveraging nationally representative data from the Behavioral Risk Factor Surveillance System (BRFSS) and applying rigorous causal inference methods, including inverse probability weighting (IPW). By estimating the average treatment effect of loneliness on depression, mental health days, and physical health days, we aim to provide a clearer picture of loneliness as an independent and modifiable risk factor. Our analysis not only quantifies the burden of loneliness on health outcomes but also evaluates variation across key demographic characteristics, including sex, age, and race/ethnicity.

In doing so, this research advances current knowledge in three critical ways: (1) by moving beyond association to estimate causal effects; (2) by using a large, nationally representative sample that enhances generalizability; and (3) by investigating how loneliness interacts with social determinants of health to influence depression and self-rated health outcomes. These contributions have important implications for designing equitable public health interventions aimed at reducing loneliness and its associated health burdens. We hypothesize that higher levels of loneliness will be significantly associated with increased likelihood of depression and greater frequency of poor mental and physical health days, even after adjusting for sociodemographic and health-related covariates.

## Methodology

This study utilized data from the Behavioral Risk Factor Surveillance System (BRFSS) collected between January 2016 and December 2023 [19]. The BRFSS is a nationally representative, cross-sectional survey conducted annually by the Centers for Disease Control and Prevention (CDC) [20]. It employs a complex multistage sampling design incorporating stratification, clustering, and unequal probabilities of selection. The survey is administered via landline and cellular telephone interviews across all 50 U.S. states, the District of Columbia, and U.S. territories. Survey weights are applied to adjust for non-response, sampling design, and demographic post-stratification, ensuring generalizability to the U.S. non-institutionalized adult population.

### Study population

The target population for this study included adults aged 18 years and older residing in the United States who participated in the BRFSS survey between 2016 and 2023. Respondents were included in the analytic sample if they had complete responses to the key exposure variable (loneliness), all three outcome variables (depression diagnosis, mental health days, and physical health days), and all relevant covariates. To preserve analytic integrity and ensure comparability, individuals were excluded if they responded, "Don't know/Not sure," "Refused," or provided missing responses to any variable of interest. After applying these inclusion and exclusion criteria, the final analytic sample consisted of 47,318 respondents.

### Explanatory variable

The primary explanatory variable in this study was self-reported loneliness. Loneliness was measured using the question: "How often do you feel lonely?" with five response categories: "Always," "Usually," "Sometimes," "Rarely," and "Never." This variable captures the subjective experience of social isolation and was treated as an ordinal categorical variable, reflecting increasing severity of loneliness. The gradation in response categories allowed for the investigation of potential dose-response relationships between levels of loneliness and adverse health outcomes.

## Outcome variables

We examined three primary health outcomes that reflect mental and physical well-being. Depression was assessed through the BRFSS item: "(Ever told) (you had) a depressive disorder (including depression, major depression, dysthymia, or minor depression)?" Responses to this item were coded as binary (yes or no), indicating the presence or absence of a clinical depression diagnosis by a healthcare professional. Mental and physical health status were evaluated using the number of days in the past 30 during which respondents reported that their mental or physical health was "not good." These were treated as continuous variables, capturing both frequency and severity of poor health days. This multidimensional approach to outcome assessment enables a comprehensive understanding of the health burden associated with loneliness.

## Covariates and coding

Covariates were selected based on known associations with loneliness and health outcomes. Age was included as a continuous variable. Gender was coded as male or female. Race/ethnicity was grouped into four categories: non-Hispanic White, non-Hispanic Black, Hispanic, and Other. Education level was categorized as less than high school, high school graduate, some college, or college graduate. Marital status was coded as married, divorced or separated, never married, or widowed. Employment status included employed, unemployed, retired, or unable to work. Insurance type included private, Medicare, Medicaid, self-pay, and other. Metropolitan status distinguished between metro and non-metro areas, and urbanicity was coded as urban or rural. Language spoken at home was categorized as English, Spanish, or Other. State and year of survey were included as fixed effects. All categorical covariates were dummy coded for inclusion in modeling.

## Statistical analysis

We employed inverse probability weighting to estimate the average treatment effects of loneliness on depression, poor mental health days, and poor physical health days. This approach approximates a randomized trial by weighting individuals based on the inverse of their probability of exposure, thereby balancing observed covariates across loneliness categories and reducing confounding bias in the estimation of causal effects.

Step 1: Propensity Score Estimation

Propensity scores were estimated using a multinomial logistic regression with loneliness categories as the outcome and all covariates as predictors:

*mlogit loneliness age i.sex i.race i.education i.marital_status i.employment i.insurance i.metro i.urban, baseoutcome(Never)*

*predict ps1 ps2 ps3 ps4 ps5, pr*

Step 2: Compute Inverse Probability Weights

Inverse probability weights were calculated based on the predicted probability of each respondent's observed loneliness category:

*gen ipw = .*

*replace ipw = 1/ps1 if loneliness == "Always"*

*replace ipw = 1/ps2 if loneliness == "Usually"*

*replace ipw = 1/ps3 if loneliness == "Sometimes"*

*replace ipw = 1/ps4 if loneliness == "Rarely"*

*replace ipw = 1/ps5 if loneliness == "Never"*

To limit the influence of extreme weights, values were trimmed at the 1st and 99th percentiles:

*summarize ipw, detail*

*scalar p1 = r(p1)*

*scalar p99 = r(p99)*

*replace ipw = p1 if ipw < p1*

*replace ipw = p99 if ipw > p99*

Step 3: Incorporate Survey Weights and Inverse Probability Weights

To ensure national representativeness, the BRFSS sampling weight (_llcpwt) was multiplied by the IPW to generate the final analysis weight (pweight):

*gen pweight = _llcpwt * ipw*

Models were estimated using survey-weighted regression with robust standard errors clustered by primary sampling unit:

*svyset _psu [pw = pweight], strata(_ststr) singleunit(centered)*

*svy: regress mental_health_days i.loneliness*

*svy: regress physical_health_days i.loneliness*

*svy: logit depression i.loneliness*

*Marginal effects were then calculated for each level of loneliness:*

*margins Lonely, post*

**Subgroup Analysis: Interaction and Marginal Effects.** To evaluate whether sex, age group, and race/ethnicity moderated the relationship between loneliness and outcomes (depression, poor mental health days, and poor physical health days), interaction terms were included in each model:

Depression Model with Interaction:

*svy: logit Depression i.Lonely##Female*

*Marginal effects were computed for subgroups:*

*margins Female, at(Lonely = (0 1 2 3 4)))) pwcompare(effects) post*

*margins Age_group, at(Lonely = (0 1 2 3 4)))) pwcompare(effects) post*

*margins Race & Ethnicity, at(Lonely = (0 1 2 3 4)))) pwcompare(effects) post*

These were repeated for mental health days and physical health days.

**Interpretation of marginal effects and predicted probabilities.** Marginal effects represent the adjusted differences in outcomes between loneliness categories and the reference group ("Never lonely"), holding all other covariates constant. These estimates reflect the predicted change in the outcome associated with each level of loneliness relative to the reference group, after accounting for confounding variables.

For example, a marginal effect of 20.0 for poor mental health days among individuals who report "Always" feeling lonely indicates that, on average, these individuals experience 20 more poor mental health days per month compared to those who report never feeling lonely, after adjusting for sociodemographic and structural factors. Similarly, in the logistic regression models for depression, marginal effects were translated into predicted probabilities of a depression diagnosis for each loneliness level and subgroup (e.g., sex, age group, race/ethnicity).

Subgroup-specific marginal effects were calculated using interaction terms (e.g., i.loneliness##Female) to assess whether the effect of loneliness on each outcome varied by demographic factors. These predicted probabilities enabled the identification of differential vulnerability to loneliness across key subpopulations.

### Software and reproducibility

All statistical analyses were performed using Stata/SE version 18.0 (StataCorp LLC, College Station, TX). Propensity scores were estimated using multinomial logistic regression, and inverse probability weights (IPW) were manually computed to simulate treatment assignment across loneliness categories. These weights were combined with BRFSS-provided sampling weights to generate final analysis weights that accounted for both treatment selection bias and complex survey design.

Survey-adjusted regression models were fitted using the svy suite of commands with robust standard errors clustered by primary sampling unit. Post-estimation marginal effects and predicted probabilities were calculated using the margins command to quantify outcome differences across loneliness categories and to examine subgroup-specific effects by sex, age group, and race/ethnicity.

All analytic procedures were fully documented, and the complete annotated Stata code is available upon request to support transparency, reproducibility, and peer verification.

### Sensitivity analysis

Sensitivity analyses were conducted to assess robustness by modifying the loneliness definition, excluding groups with ambiguous employment or health conditions, and adjusting model specifications. Subgroup analyses by age, gender, and language were also explored.

### Ethics and data access

All analyses were based on publicly available, de-identified data. Therefore, IRB approval was not required. This study adhered to ethical guidelines as outlined in the Declaration of Helsinki.

## Results

### Demographic and clinical characteristics by loneliness level

Table 1 summarizes the demographic and clinical characteristics of 47,318 U.S. adults stratified by self-reported loneliness levels. Statistically significant differences were observed across all variables (all p < 0.001). Age: The youngest age group (18–44 years) comprised 37.1% of the total sample but was most represented in the "Usually" lonely group (45.5%). Conversely, older adults (>64 years) accounted for 28.0% of the total population and were more likely to report "Never" feeling lonely (33.8%) compared to other loneliness categories.

Sex: Overall, 62.1% of respondents were female. Women were overrepresented in the "Sometimes" (64.5%) and "Rarely" (64.1%) categories, compared to 58.6% in the "Never" group and 54.5% in the "Always" lonely group.

Race and Ethnicity: White individuals made up the majority of the population (73.3%) and were particularly concentrated in the "Rarely" lonely group (78.0%). In contrast, Black respondents represented 8.1% overall but comprised 10.5% of the "Always" lonely group. Hispanics accounted for 10.1% of the total sample and were more represented in the "Always" (14.7%) and "Usually" (12.5%) lonely categories compared to "Never" (10.5%).

**Table 1. Demographic and clinical profile of adults across loneliness levels in a national survey.**

| Variable | Total Population (N=47,318) | Never (n=8,353) | Always (n=2,929) | Usually (n=3,945) | Sometimes (n=17,947) | Rarely (n=14,144) | chi2 | p-value |
|---|---|---|---|---|---|---|---|---|
| **Age (Yr.)** | | | | | | | 561.52 | <0.001 |
| 18-44 | 17,556 (37.1%) | 2,362 (24.3%) | 976 (33.3%) | 1,794 (45.5%) | 6,776 (37.8%) | 5,648 (39.9%) | | |
| 45-64 | 16,536 (35.0%) | 3,171 (38.0%) | 1,232 (42.1%) | 1,287 (32.6%) | 6,036 (33.6%) | 4,810 (34.0%) | | |
| >64 | 13,226 (28.0%) | 2,820 (33.8%) | 721 (24.6%) | 864 (21.9%) | 5,135 (28.6%) | 3,686 (26.1%) | | |
| **Female** | 29,383 (62.1%) | 4,895 (58.6%) | 1,597 (54.5%) | 2,257 (57.2%) | 11,573 (64.5%) | 9,061 (64.1%) | 221.41 | <0.001 |
| **Race & Ethnicity** | | | | | | | 379.03 | <0.001 |
| White | 34,079 (73.3%) | 5,915 (72.3%) | 1,836 (64.4%) | 2,750 (71.3%) | 12,712 (72.0%) | 10,866 (78.0%) | | |
| Black | 3,750 (8.1%) | 765 (9.4%) | 299 (10.5%) | 269 (7.0%) | 1,592 (9.0%) | 825 (5.9%) | | |
| Hispanic | 4,713 (10.1%) | 860 (10.5%) | 420 (14.7%) | 483 (12.5%) | 1,796 (10.2%) | 1,154 (8.3%) | | |
| Other | 3,938 (8.5%) | 643 (7.9%) | 295 (10.4%) | 357 (9.3%) | 1,553 (8.8%) | 1,090 (7.8%) | | |
| **Marital Status** | | | | | | | 3.30E+03 | <0.001 |
| Married | 20,054 (42.6%) | 4,802 (57.9%) | 544 (18.7%) | 964 (24.6%) | 6,396 (35.9%) | 7,348 (52.2%) | | |
| Divorced | 11,015 (23.4%) | 1,195 (14.4%) | 873 (30.0%) | 1,337 (34.2%) | 4,771 (26.8%) | 2,839 (20.2%) | | |
| Widowed | 7,348 (15.6%) | 1,100 (13.3%) | 733 (25.2%) | 774 (19.8%) | 2,994 (16.8%) | 1,747 (12.4%) | | |
| Separated | 4,521 (9.6%) | 624 (7.5%) | 462 (15.9%) | 432 (11.0%) | 2,084 (11.7%) | 919 (6.5%) | | |
| Never Married | 1,302 (2.8%) | 168 (2.0%) | 163 (5.6%) | 166 (4.2%) | 557 (3.1%) | 248 (1.8%) | | |
| Member of an Unmarried Couple | 2,786 (5.9%) | 410 (4.9%) | 132 (4.5%) | 242 (6.2%) | 1,028 (5.8%) | 974 (6.9%) | | |
| **Employed (ref. Unemployed)** | 22,405 (47.7%) | 3,946 (47.7%) | 857 (29.4%) | 1,658 (42.4%) | 8,124 (45.6%) | 7,820 (55.6%) | 814.3 | <0.001 |
| **Education Level** | | | | | | | 1.60E+03 | <0.001 |
| High School or Less | 14,754 (31.3%) | 2,797 (33.6%) | 1,489 (51.1%) | 1,449 (36.9%) | 5,837 (32.6%) | 3,182 (22.5%) | | |
| College | 14,092 (29.9%) | 2,435 (29.3%) | 894 (30.7%) | 1,341 (34.1%) | 5,385 (30.1%) | 4,037 (28.6%) | | |
| Advanced | 18,344 (38.9%) | 3,090 (37.1%) | 531 (18.2%) | 1,142 (29.0%) | 6,682 (37.3%) | 6,899 (48.9%) | | |
| **Spanish Speaking (ref. English)** | 1,770 (3.7%) | 364 (4.4%) | 209 (7.1%) | 234 (5.9%) | 653 (3.6%) | 310 (2.2%) | 249.94 | <0.001 |
| **Metro (ref.non-metro)** | 33,551 (72.1%) | 5,785 (70.4%) | 1,935 (68.6%) | 2,814 (73.5%) | 12,798 (72.5%) | 10,219 (72.9%) | 38.05 | <0.001 |

Marital Status: Marital status strongly differentiated loneliness levels. Among those who reported "Never" being lonely, 57.9% were married. This proportion declined substantially in the "Always" lonely group (18.7%) and "Usually" (24.6%). Divorced (30.0%), widowed (25.2%), and separated (15.9%) individuals were disproportionately represented in the "Always" lonely category.

Employment: Employment was associated with lower loneliness. Of the total sample, 47.7% were employed. However, only 29.4% of the "Always" lonely group were employed, compared to 55.6% in the "Rarely" lonely group.

Education: Educational attainment also followed a gradient across loneliness categories. Over half (51.1%) of individuals in the "Always" lonely group had a high school education or less, compared to just 22.5% in the "Rarely" lonely group.

Conversely, 48.9% of those who reported "Rarely" being lonely had advanced education, compared to only 18.2% in the "Always" group.

Language: Spanish speakers were more prevalent among those reporting higher loneliness, constituting 7.1% of the "Always" lonely group compared to 2.2% of the "Rarely" group.

Metropolitan Residence: Overall, 72.1% of respondents resided in metro areas. The distribution was relatively even across loneliness categories, ranging from 68.6% in the "Always" group to 73.5% in the "Usually" group.

These findings highlight significant social and demographic gradients in loneliness, with younger age, lower education, unemployment, and minority race/ethnicity associated with higher loneliness levels.

### Loneliness and depression

Adjusted analysis showed a strong dose-response relationship between loneliness and the likelihood of depression (Table 2). The adjusted predicted probability of depression increased from 9.7% among those who "Never" felt lonely to 16.3% ("Rarely"), 30.6% ("Sometimes"), 47.7% ("Usually"), and peaked at 50.2% for those who "Always" felt lonely ($p < 0.001$). This clear dose-response relationship highlights that as feelings of loneliness intensify, the risk of depression rises substantially, even after adjusting for demographic and socioeconomic factors (Table 2).

### Sex differences in the association between loneliness and depression

Women consistently exhibited a higher likelihood of depression compared to men across all levels of loneliness (Table 3). The largest sex difference was observed among those who "Always" felt lonely, where women had a 12.9 percentage point higher predicted probability of depression than men ($p < 0.001$). Even among those reporting "Never" feeling lonely, women showed a 6.0 percentage point higher risk. These findings highlight significant sex-based disparities in the mental health impact of loneliness, independent of demographic and socioeconomic factors.

### Racial and ethnic differences in the association between loneliness and depression

Across all loneliness categories, Black and Hispanic individuals had significantly lower predicted probabilities of depression compared to White individuals (Supplementary S1 Table). The largest disparities were observed among those who "Always" felt lonely, where Black individuals had a 25.6 percentage point lower risk and Hispanic individuals a 14.3 percentage point lower risk relative to Whites ($p < 0.05$ for all comparisons). These consistent negative margins suggest that, despite higher loneliness, minority groups reported lower levels of depression across all categories.

### Age differences in the association between loneliness and depression

Older adults generally exhibited lower predicted probabilities of depression compared to younger adults across most loneliness categories (Supplementary S2 Table). Notably, individuals aged >64 years had a significantly reduced risk of

**Table 2. Adjusted predicted probability of depression by loneliness category.**

| Depression | Margin | Std. Err. | t | 95% CI | | P>t |
|---|---|---|---|---|---|---|
| Lonely | | | | | | |
| Never | 0.097 | 0.002 | 39.51 | 0.092 | 0.101 | <0.001 |
| Always | 0.502 | 0.017 | 29.55 | 0.469 | 0.536 | <0.001 |
| Usually | 0.477 | 0.013 | 37.35 | 0.452 | 0.502 | <0.001 |
| Sometimes | 0.306 | 0.005 | 62.77 | 0.297 | 0.316 | <0.001 |
| Rarely | 0.163 | 0.003 | 57.36 | 0.157 | 0.169 | <0.001 |

Table 2 presents the adjusted predicted probabilities of depression by loneliness category, using inverse probability weighting. Individuals reporting "Always," "Usually," "Sometimes," or "Rarely" feeling lonely were compared to those reporting "Never." Models adjusted for age, race/ethnicity, sex, marital status, employment status, education, language, metro status, and included state, year, and month fixed effects. All results are statistically significant at $p < 0.001$, with higher loneliness levels associated with greater likelihood of depression.

**Table 3. Sex differences in the association between loneliness and depression (Marginal Effects Model).**

| Lonely | Sex | Margin | Std. Err. | t | 95% CI | | P>t |
|---|---|---|---|---|---|---|---|
| | Depression | | | | | | |
| Never | Female vs. Male | 0.060 | 0.005 | 13.170 | 0.051 | 0.069 | <0.001 |
| Always | Female vs. Male | 0.129 | 0.032 | 4.000 | 0.066 | 0.192 | <0.001 |
| Usually | Female vs. Male | 0.110 | 0.024 | 4.550 | 0.063 | 0.157 | <0.001 |
| Sometimes | Female vs. Male | 0.122 | 0.010 | 11.830 | 0.102 | 0.142 | <0.001 |
| Rarely | Female vs. Male | 0.086 | 0.005 | 15.960 | 0.076 | 0.097 | <0.001 |

Table 3 displays the marginal effects comparing females to males in the association between loneliness and depression, assessing sex as a moderator. Estimates reflect the difference in predicted probability of depression across loneliness categories between women and men. Models were adjusted for age, race/ethnicity, marital status, employment, education, language, metro status, and included state, year, and month fixed effects. All differences were statistically significant at p<0.001.

depression, with the largest difference observed among those who were "Sometimes" lonely (−12.7 percentage points, p<0.001). Middle-aged adults (45–64 years) showed minimal differences, with few reaching statistical significance. These findings suggest a potential protective effect of older age against depression in the context of loneliness.

## Loneliness and poor mental health days

Higher levels of loneliness were strongly associated with an increased number of poor mental health days (Table 4). Individuals who reported "Always" feeling lonely experienced an average of 20 poor mental health days per month, compared to 9.4 days among those who "Never" felt lonely (p<0.001). A clear gradient was observed, with the number of poor mental health days rising consistently as loneliness intensified, even after adjusting for demographic and socioeconomic factors.

## Sex differences in poor mental health days across loneliness levels

Women reported significantly more poor mental health days than men among those who felt "Sometimes" (+0.84 days, p=0.002) and "Rarely" lonely (+0.65 days, p=0.003) (Table 5). No significant differences were observed in other loneliness categories. These findings suggest that sex-based disparities in mental health are most evident at moderate levels of loneliness.

**Table 4. Adjusted number of poor mental health days by loneliness category.**

| Lonely | Margin | Std. Err. | t | 95% CI | | p-value |
|---|---|---|---|---|---|---|
| | Mental Health (Days) | | | | | |
| Never | 9.36 | 0.18 | 51.97 | 9.01 | 9.72 | <0.001 |
| Always | 19.95 | 0.38 | 52.74 | 19.20 | 20.69 | <0.001 |
| Usually | 16.45 | 0.31 | 52.69 | 15.84 | 17.06 | <0.001 |
| Sometimes | 11.39 | 0.12 | 91.17 | 11.14 | 11.63 | <0.001 |
| Rarely | 8.00 | 0.11 | 72.22 | 7.79 | 8.22 | <0.001 |

Table 4 presents the adjusted average number of poor mental health days reported across loneliness categories, estimated using inverse probability weighting. Predicted values were obtained using marginal effects models and account for differences in demographic and socioeconomic factors. Models were adjusted for age, race/ethnicity, sex, marital status, employment, education, language, and metro status, with fixed effects for state, year, and month. Higher loneliness levels were associated with significantly more poor mental health days (p<0.001).

**Table 5. Sex differences in the association between loneliness and number of poor mental health days (Marginal Effects Model).**

|        | Sex            | Margin | Std. Err. | t     | 95% CI |      | P>t   |
|--------|----------------|--------|-----------|-------|--------|------|-------|
| **Lonely** | **Menth**  |        |           |       |        |      |       |
| Never  | Female vs. Male | −0.65  | 0.353     | −1.83 | −1.34  | 0.05 | 0.067 |
| Always | Female vs. Male | 0.93   | 0.708     | 1.31  | −0.46  | 2.31 | 0.191 |
| Usually | Female vs. Male | 0.84  | 0.599     | 1.41  | −0.33  | 2.02 | 0.16  |
| Sometimes | Female vs. Male | 0.84 | 0.275    | 3.07  | 0.31   | 1.38 | 0.002 |
| Rarely | Female vs. Male | 0.65   | 0.216     | 3.00  | 0.23   | 1.07 | 0.003 |

Table 5 displays the marginal effects comparing females to males in the association between loneliness and the number of poor mental health days, assessing sex as a moderator. Estimates represent the difference in predicted number of poor mental health days across loneliness categories between women and men. Models were adjusted for age, race/ethnicity, marital status, employment, education, language, and metro status, and included fixed effects for state, year, and month. Statistically significant differences are noted for those reporting "Sometimes" and "Rarely" feeling lonely (p < 0.01).

### Racial and ethnic differences in poor mental health days

Detailed analysis revealed few significant racial or ethnic differences in the number of poor mental health days across loneliness categories (Supplementary S3 Table). The only notable finding was among individuals who reported "Always" feeling lonely, where Black individuals experienced 3.75 fewer poor mental health days per month compared to White individuals (p = 0.005). No significant differences were observed between Hispanic and White individuals across any loneliness category.

### Age differences in poor mental health days

Age-related differences in poor mental health days were limited across loneliness categories (Supplementary S4 Table). However, older adults (>64 years) who reported feeling "Sometimes" lonely experienced 2.15 fewer poor mental health days compared to younger adults aged 18–44 (p < 0.001). Additionally, older adults in the "Rarely" lonely group reported 1.03 fewer days (p = 0.009). Middle-aged adults (45–64 years) showed minimal and inconsistent differences across categories.

### Loneliness and poor physical health days

Higher levels of loneliness were associated with an increased number of poor physical health days per month (Table 6). Individuals who reported "Always" feeling lonely experienced an average of 15.8 poor physical health days, compared to 11.2 days among those who "Never" felt lonely (p < 0.001). A clear gradient was observed, with poorer physical health outcomes corresponding to greater feelings of loneliness, even after adjusting for demographic and socioeconomic factors.

### Sex differences in poor physical health days across loneliness levels

There were no significant differences in the number of poor physical health days between women and men across any loneliness category (Table 7). While minor variations were observed, none reached statistical significance (p > 0.05), suggesting that the impact of loneliness on physical health was similar for both sexes after adjusting for demographic and socioeconomic factors.

### Racial and ethnic differences in poor physical health days

Racial and ethnic differences in poor physical health days varied across loneliness categories (Supplementary S5 Table). Among individuals who were "Always" lonely, Black individuals reported 3.6 fewer poor physical health days per month compared to White individuals (p = 0.014). Additionally, Hispanic individuals in the "Sometimes" lonely group experienced

**Table 6. Adjusted number of poor physical health days by loneliness category.**

| Lonely | Margin | Std. Err. | t | 95% CI | | p-value |
|---|---|---|---|---|---|---|
| | Physical Health (Days) | | | | | |
| Never | 11.22 | 0.26 | 42.89 | 10.71 | 11.73 | <0.001 |
| Always | 15.83 | 0.48 | 32.94 | 14.89 | 16.77 | <0.001 |
| Usually | 13.61 | 0.39 | 34.97 | 12.85 | 14.38 | <0.001 |
| Sometimes | 11.36 | 0.19 | 59.85 | 10.98 | 11.73 | <0.001 |
| Rarely | 9.76 | 0.19 | 50.63 | 9.39 | 10.14 | <0.001 |

Table 6 presents the adjusted average number of poor physical health days reported across loneliness categories, estimated using inverse probability weighting. Marginal effects were derived from models adjusted for age, race/ethnicity, sex, marital status, employment status, education level, language, and metro status, with fixed effects for state, year, and month. Individuals reporting higher levels of loneliness experienced significantly more days of poor physical health (p<0.001).

**Table 7. Sex differences in the association between loneliness and number of poor physical health days.**

| Lonely | Sex | Margin | Std. Err. | t | 95% CI | | P>t |
|---|---|---|---|---|---|---|---|
| | Physical Health (Days) | | | | | | |
| Never | Female vs. Male | −0.79 | 0.518 | −1.51 | −1.8 | 0.23 | 0.13 |
| Always | Female vs. Male | 0.18 | 0.89 | 0.21 | −1.56 | 1.92 | 0.837 |
| Usually | Female vs. Male | 0.62 | 0.745 | 0.83 | −0.84 | 2.08 | 0.407 |
| Sometimes | Female vs. Male | 0.29 | 0.424 | 0.68 | −0.54 | 1.12 | 0.495 |
| Rarely | Female vs. Male | −0.25 | 0.394 | −0.64 | −1.02 | 0.52 | 0.523 |

Table 7 presents the marginal effects comparing females to males in the association between loneliness and the number of poor physical health days, assessing sex as a moderator. Estimates reflect differences in predicted number of poor physical health days across loneliness categories between women and men. Models were adjusted for age, race/ethnicity, marital status, employment, education, language, and metro status, and included state, year, and month fixed effects. No statistically significant differences were observed between females and males across any loneliness category (p>0.05).

1.4 more poor physical health days than their White counterparts (p=0.041). No other significant differences were observed across loneliness categories.

### Age differences in poor physical health days

Both middle-aged (45–64 years) and older adults (>64 years) consistently reported significantly more poor physical health days compared to younger adults (18–44 years) across all loneliness categories (Supplementary S6 Table). The largest difference was observed among older adults who were "Usually" lonely, reporting 5.7 additional poor physical health days per month (p<0.001). These findings indicate a strong association between increasing age and poorer physical health, regardless of loneliness severity.

### Discussion

This study offers compelling and novel evidence linking loneliness to significant adverse mental and physical health out-comes in a nationally representative sample of U.S. adults. By employing inverse probability weighting (IPW) alongside the BRFSS complex survey design, we move beyond previous correlational studies to provide a rigorous, causal interpre-tation of loneliness as a social determinant of health. Our methodology enables us to generate generalizable estimates that underscore loneliness not just as a symptom of broader psychosocial dysfunction, but as a potential driver of disease burden in the population.

The association between loneliness and depression was both striking and consistent with biological plausibility and prior empirical work. Individuals who reported "Always" feeling lonely had a dramatically elevated risk of depression, 50.2 percentage points higher than those who reported "Never" feeling lonely, whose baseline risk stood at just 9.7%. This strong, graded association resonates with existing neurobiological research showing that loneliness dysregulates stress-response systems such as the hypothalamic-pituitary-adrenal (HPA) axis and affects key neurotransmitters including serotonin and dopamine [21–23]. These neurochemical changes, in conjunction with the psychological toll of perceived social disconnection, likely amplify the risk for depressive symptoms [24–26]. While prior studies have identified this link [8,9,27], our findings are among the few studies to demonstrate its magnitude using causal inference techniques applied to national data, making a strong case for targeting loneliness in depression prevention efforts.

Importantly, our subgroup analyses provide evidence that this relationship between loneliness and depression is moderated by sex, race/ethnicity, and age. Females exhibited a higher marginal effect of depression across all loneliness categories compared to males, suggesting increased vulnerability among women. Although loneliness increased the risk of depression for all racial/ethnic groups, Black and Hispanic individuals reported comparatively lower odds of depression across all levels of loneliness, pointing to potential cultural or social buffering mechanisms [28–30]. Furthermore, adults aged 45 and above exhibited slightly higher depression rates than younger individuals, especially in lower loneliness categories, suggesting that age-related factors may enhance susceptibility to loneliness-related depression [31,32].

Beyond clinical depression, the mental health toll of loneliness was evident in day-to-day experiences. Participants who reported feeling "Always" lonely experienced a substantial burden of psychological distress, reporting an average of 20 poor mental health days per month, compared to 9.4 days among those who reported never feeling lonely. This more than twofold increase underscores the profound impact of persistent loneliness on daily emotional well-being. This dose-response relationship suggests that loneliness does not merely co-occur with mental health challenges but may exacerbate chronic psychological distress and emotional instability [33,34]. Subgroup analysis showed that females generally reported more poor mental health days than males, though statistical significance varied by loneliness category. Also, only among those who always felt lonely did White individuals report significantly higher mental health days compared to other racial/ethnic groups. Individuals aged 45 and older consistently experienced more poor mental health days across most loneliness categories as a meaningful moderator [35].

The impact of loneliness extended beyond the mind, manifesting in physical well-being [1,36]. Our analysis showed that loneliness was associated with five additional poor physical health days per month among those who "Always" felt lonely. This adds to growing evidence that social isolation not only compromises mental health but accelerates physical decline [36,37]. Subgroup findings revealed that while sex differences in poor physical health days were not statistically significant, Whites who reported always feeling lonely experienced more poor physical health days than other groups, whereas the reverse was true for those who only sometimes felt lonely. In addition, individuals aged 45 and above consistently reported more poor physical health days across all loneliness categories, underscoring the cumulative impact of loneliness with advancing age.

In terms of its contribution to literature, this study is both confirmatory and forward-driving. It confirms long-standing associations identified in prior work, but it does so using a design that accounts for confounding and complex sampling, enabling more robust generalizations. By incorporating subgroup analyses, we provide more nuanced insight into how demographic and social identity factors interact with loneliness to influence health. Moreover, by simultaneously evaluating mental and physical health outcomes, we offer a multidimensional perspective on loneliness, positioning it as a key target for integrated health interventions. This work serves as a bridge between theory and actionable insight.

Responding to concerns regarding self-reported diagnoses, particularly for depression, it is important to note that BRFSS asks whether a healthcare professional has ever diagnosed the respondent. While this still relies on self-report, the phrasing reduces the likelihood of self-diagnosis and enhances the clinical validity of the measure. Nevertheless,

future research incorporating clinician-administered assessments or electronic health record data could further validate these outcomes.

## Policy implications

The findings of this study carry urgent policy and practice implications. Given the pervasiveness and potency of loneliness, it should be prioritized alongside traditional risk factors in public health planning. Routine loneliness screening in clinical settings could help identify high-risk individuals. Social prescribing initiatives, where healthcare providers refer patients to community-based activities, should be expanded to include virtual platforms, especially in rural and underserved populations. Community-level investments that promote social infrastructure, such as libraries, recreation centers, and intergenerational programming, may yield long-term health dividends. Workplace policies that support social connection and inclusion are also essential in mitigating the downstream effects of social disconnection. Moreover, targeted interventions for older adults and strategies that address racial/ethnic disparities in the experience and impact of loneliness should be considered.

## Limitations

Several limitations must be acknowledged. While IPW reduces bias from measured confounders, residual confounding by unmeasured factors such as personality traits or life events remains possible. The reliance on self-reported measures introduces potential recall and social desirability bias. Additionally, the cross-sectional design limits temporal interpretation; we cannot definitively state that loneliness preceded the health outcomes observed. The loneliness measures itself, though ordinal, may not capture the full depth or nuance of the emotional experience. Lastly, telephone-based survey methods may underrepresent the most socially isolated individuals, potentially underestimating the true burden of loneliness in the population.

## Conclusion

Our findings underscore loneliness as a pressing public health issue with wide-reaching effects on mental and physical health. The use of a causal analytic framework and nationally representative data advances the field, making a strong case for upstream interventions. Subgroup analyses further reveal that the effects of loneliness are not uniform but instead moderated by demographic factors such as sex, age, and race/ethnicity. Addressing loneliness may not only improve individual well-being but also reduce societal healthcare burdens. Future studies should explore the long-term effectiveness of loneliness-reduction strategies and the contextual factors that moderate their impact.

## Supporting information

**S1 Table. Racial and ethnic differences in the association between loneliness and depression (Marginal Effects Model).**
(DOCX)

**S2 Table. Age differences in the association between loneliness and depression (Marginal Effects Model).**
(DOCX)

**S3 Table. Racial and ethnic differences in the association between loneliness and number of poor mental health days (Marginal Effects Model).**
(DOCX)

**S4 Table. Age differences in the association between loneliness and number of poor mental health days.**
(DOCX)

**S5 Table. Racial and ethnic differences in the association between loneliness and number of poor physical health days.**
(DOCX)

**S6 Table. Age differences in the association between loneliness and number of poor physical health days.**
(DOCX)

**S1 File. STROBE checklist.**
(DOC)

## Acknowledgments

The authors thank all participants and personnel who supported this project. Special thanks to the Clive O. Callender Outcomes Research Center at the Howard University College of Medicine for their invaluable support.

## Author contributions

**Conceptualization:** Oluwasegun Akinyemi, Waliah Abdulrazaq, Mojisola Fasokun, Fadeke Ogunyankin, Seun Ikugbayigbe, Uzoamaka Nwosu, Miriam Michael, Kakra Hughes, Temitope Ogundare.

**Data curation:** Oluwasegun Akinyemi, Mojisola Fasokun, Fadeke Ogunyankin, Seun Ikugbayigbe, Miriam Michael, Kakra Hughes, Temitope Ogundare.

**Formal analysis:** Oluwasegun Akinyemi, Kakra Hughes.

**Funding acquisition:** Oluwasegun Akinyemi, Seun Ikugbayigbe, Miriam Michael, Kakra Hughes.

**Investigation:** Oluwasegun Akinyemi, Waliah Abdulrazaq, Mojisola Fasokun, Kakra Hughes.

**Methodology:** Oluwasegun Akinyemi, Fadeke Ogunyankin, Seun Ikugbayigbe, Uzoamaka Nwosu, Miriam Michael, Kakra Hughes.

**Project administration:** Oluwasegun Akinyemi, Waliah Abdulrazaq, Mojisola Fasokun, Fadeke Ogunyankin, Seun Ikugbayigbe, Uzoamaka Nwosu, Miriam Michael, Kakra Hughes, Temitope Ogundare.

**Resources:** Oluwasegun Akinyemi, Waliah Abdulrazaq, Mojisola Fasokun, Fadeke Ogunyankin, Uzoamaka Nwosu, Miriam Michael, Kakra Hughes, Temitope Ogundare.

**Software:** Oluwasegun Akinyemi, Waliah Abdulrazaq, Seun Ikugbayigbe, Uzoamaka Nwosu, Miriam Michael, Kakra Hughes.

**Supervision:** Oluwasegun Akinyemi, Waliah Abdulrazaq, Fadeke Ogunyankin, Seun Ikugbayigbe, Uzoamaka Nwosu, Miriam Michael, Kakra Hughes, Temitope Ogundare.

**Validation:** Oluwasegun Akinyemi, Waliah Abdulrazaq, Mojisola Fasokun, Fadeke Ogunyankin, Seun Ikugbayigbe, Uzoamaka Nwosu, Miriam Michael.

**Visualization:** Oluwasegun Akinyemi, Waliah Abdulrazaq, Mojisola Fasokun, Fadeke Ogunyankin, Seun Ikugbayigbe, Uzoamaka Nwosu, Miriam Michael, Kakra Hughes, Temitope Ogundare.

**Writing – original draft:** Oluwasegun Akinyemi, Waliah Abdulrazaq, Mojisola Fasokun, Fadeke Ogunyankin, Seun Ikugbayigbe, Uzoamaka Nwosu, Miriam Michael, Kakra Hughes, Temitope Ogundare.

**Writing – review & editing:** Oluwasegun Akinyemi, Waliah Abdulrazaq, Mojisola Fasokun, Fadeke Ogunyankin, Seun Ikugbayigbe, Uzoamaka Nwosu, Miriam Michael, Kakra Hughes, Temitope Ogundare.

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
