## [Decision Letter · Decision Letter 0]

May 08 2025

PONE-D-25-04971The Impact of Loneliness on Depression, Mental Health Days, and Physical HealthPLOS ONE

Dear Dr. Akinyemi,

Thank you for submitting your manuscript to PLOS ONE. After careful consideration, we feel that it has merit but does not fully meet PLOS ONE’s publication criteria as it currently stands. Therefore, we invite you to submit a revised version of the manuscript that addresses the points raised during the review process.

**ACADEMIC EDITOR: **

 This study investigates the associations of loneliness with depression, mental health, and physical health, accounting for confounders and state and year fixed effects. Data analysis used Inverse Probability Weighting (IPW) to estimate the Average Treatment Effect (ATE) of loneliness on depression, mental health days, and physical health days. Findings suggest loneliness as a strong predictor of depression and poor mental and physical health. While this method was used to minimize selection bias, please include some rationales underlying the use of matched and unmatched groups in the study methods.Per the RECORD checklist, please provide information on the data cleaning methods used in the study.

We look forward to receiving your revised manuscript.

Kind regards,

Renata Komalasari

Academic Editor

PLOS ONE

Journal Requirements:

This project was supported (in part) by the National Institute on Minority Health and Health Disparities of the National Institutes of Health under Award Number 2U54MD007597. The content is solely the responsibility of the authors and does not necessarily represent the official views of the National Institutes of Health.

3. Please amend your manuscript to include your abstract after the title page.

4. Please include a copy of Table 4 which you refer to in your text on page 15 and 16.

5. Please remove all personal information, ensure that the data shared are in accordance with participant consent, and re-upload a fully anonymized data set.

7. Please address comments from each reviewer.

Reviewers' comments:

Reviewer's Responses to Questions

**Comments to the Author**

1. Is the manuscript technically sound, and do the data support the conclusions?

Reviewer #1: Partly

Reviewer #2: Partly

2. Has the statistical analysis been performed appropriately and rigorously? 

Reviewer #1: I Don't Know

Reviewer #2: Yes

3. Have the authors made all data underlying the findings in their manuscript fully available?

Reviewer #1: No

Reviewer #2: Yes

4. Is the manuscript presented in an intelligible fashion and written in standard English?

Reviewer #1: Yes

Reviewer #2: Yes

5. Review Comments to the Author

Reviewer #1: Thank you for the opportunity to review this manuscript. This study leverages data from the Behavioral Risk Factor Surveillance System (BRFSS) (2016–2023) to examine the relationship between loneliness and mental and physical health outcomes. The findings indicate that loneliness is significantly associated with an increased likelihood of depression diagnosis (+39.3 percentage points), as well as a greater number of poor mental (+10.9 days) and physical health days (+5.0 days). These results emphasize the urgent need for interventions addressing social isolation to improve population health.

I appreciate the thorough approach taken by the authors and offer the following comments for consideration:

The introduction effectively outlines the importance of studying loneliness and clearly defines the key health outcomes of interest. Please expand on the limitations of previous studies and how this study advances current knowledge. What gaps in the literature does this work specifically address? You note that social determinants may influence the relationship between loneliness and health outcomes, yet they are treated solely as covariates rather than moderators. Although subgroup analyses for race and ethnicity are included, a more granular exploration of how specific socioeconomic and structural factors contribute to disparities could provide additional insights.

Regarding the methods, please describe the inclusion and exclusion criteria for BRFSS participants. Additionally, clarify how the survey design ensures national representativeness—while this may be well-known to BRFSS users, including it will enhance accessibility for broader audiences. Further detail on how covariates were coded (e.g., categorical vs. continuous) would strengthen transparency in the methodology section.

The statistical methods appear rigorous and appropriate, particularly the use of robust standard errors and the treatment of outcomes as ordinal variables. However, additional details are needed to enhance clarity and reproducibility. Please provide stepwise equations for IPW estimation, including the propensity score model, how inverse probabilities were computed, and the final weighting approach. How were sampling weights from BRFSS incorporated into the IPW procedure? Were they applied multiplicatively? Did you implement strategies to address extreme propensity scores? Many studies cap weights at reasonable thresholds—please clarify your approach. Additional details on how the ATE was computed and what the reported values represent in the context of loneliness and health outcomes would be helpful. Would you be open to providing the statistical code (e.g., in a GitHub repository or as supplementary material)? This would significantly enhance transparency and reproducibility. Please specify the statistical software and functions/packages used for the analysis.

A useful reference providing sufficient statistical detail on similar methods is:

Morrish, N., Mujica-Mota, R., & Medina-Lara, A. (2022). Understanding the effect of loneliness on unemployment: propensity score matching. BMC Public Health, 22(1), 740. https://doi.org/10.1186/s12889-022-13107-x

The discussion effectively contextualizes the findings, but the contribution relative to prior studies could be made more explicit. Currently, the results seem to parallel previous research rather than extending it. Please clarify how your study improves upon prior work—for example, through a more robust statistical approach, use of nationally representative data, or exploration of new potential mechanisms linking loneliness to health. Strengthening the discussion on mechanisms and policy implications would further enhance its contribution.

Reviewer #2: Thank you for the opportunity to review this study.

The study's topic is important to address a common mental health risk, but there are some observations:

how was self-diagnosis avoided regarding depression. was there a mechanism used to ensure that data collected regarding diagnosis is accurate? relying on self-reported assessment for mental health makes it hard to take collected data as reliable.

6. PLOS authors have the option to publish the peer review history of their article (what does this mean? ). If published, this will include your full peer review and any attached files.

**Do you want your identity to be public for this peer review?** For information about this choice, including consent withdrawal, please see our Privacy Policy .

Reviewer #1: **Yes: ** Tyler Bell

Reviewer #2: No

---

## [Author Response · Author response to Decision Letter 1]

24 Apr 2025

Response to Academic Editor

Thank you for your thoughtful feedback. We appreciate the opportunity to clarify key methodological aspects of our study.

Academic Editor:

Comment :

This study investigates the associations of loneliness with depression, mental health, and physical health, accounting for confounders and state and year fixed effects. Data analysis used Inverse Probability Weighting (IPW) to estimate the Average Treatment Effect (ATE) of loneliness on depression, mental health days, and physical health days. Findings suggest loneliness as a strong predictor of depression and poor mental and physical health. While this method was used to minimize selection bias, please include some rationales underlying the use of matched and unmatched groups in the study methods.

Per the RECORD checklist, please provide information on the data cleaning methods used in the study.

Rationale for Use of Matched and Unmatched Groups in the Study Methods

In this study, we applied Inverse Probability Weighting (IPW) rather than traditional propensity score matching. The rationale for using IPW lies in its ability to retain the full analytic sample (unmatched approach) while minimizing selection bias due to observed confounders. Unlike matching methods, which exclude unmatched individuals and potentially reduce generalizability, IPW allows for adjustment across all levels of loneliness by weighting individuals inversely proportional to their probability of being in a given loneliness category.

This approach was particularly suited to our objective of estimating the Average Treatment Effect (ATE) across a multi-level exposure (five categories of loneliness) within a nationally representative dataset. The use of IPW, combined with BRFSS sampling weights and state/year fixed effects, enabled us to balance covariates across groups without sacrificing sample size or representativeness.

We have clarified this rationale in the Statistical Analysis section of the manuscript to explicitly distinguish why a weighting strategy was preferred over matching techniques.

Data Cleaning Methods (RECORD Checklist Compliance)

In adherence to the RECORD guidelines, we have expanded the Methodology section to detail our data cleaning procedures:

We conducted initial data integrity checks to identify incomplete or inconsistent responses.

Participants were excluded if they provided responses such as "Don’t know," "Refused," or had missing data for key variables, including the exposure (loneliness), outcomes (depression diagnosis, poor mental health days, poor physical health days), and covariates.

We verified coding consistency across survey years (2016–2023) to ensure harmonization of variable definitions.

Extreme values for continuous variables (e.g., health days) were assessed, and implausible entries outside the BRFSS-defined ranges were removed.

For the IPW procedure, we trimmed extreme weights at the 1st and 99th percentiles to mitigate undue influence from outliers.

All cleaning steps were performed using Stata, with code documentation available upon request to ensure transparency and reproducibility.

We have updated the manuscript to explicitly describe these steps under a new subsection titled Data Cleaning and Preparation within the Methodology section.

We trust that these clarifications address your concerns. We remain committed to ensuring methodological transparency and adherence to best practices.

Response to Reviewers

We are grateful for the thoughtful and constructive feedback provided by both reviewers. We have carefully considered each comment and revised the manuscript accordingly to strengthen clarity, methodological transparency, and overall contribution. Below we provide a point-by-point response detailing how each comment has been addressed.

Reviewer #1

Comment: “The introduction effectively outlines the importance of studying loneliness and clearly defines the key health outcomes of interest. Please expand on the limitations of previous studies and how this study advances current knowledge. What gaps in the literature does this work specifically address?”

Response:

We appreciate this suggestion and have revised the introduction to explicitly highlight the limitations of previous research, including over-reliance on correlational designs, use of non-representative samples, and insufficient attention to the interplay between loneliness and social determinants of health. We now clarify how our study addresses these gaps through the use of a large, nationally representative dataset (BRFSS), application of inverse probability weighting to approximate causal inference, and exploration of demographic variations. These revisions can be found in the revised Introduction section.

Comment: “You note that social determinants may influence the relationship between loneliness and health outcomes, yet they are treated solely as covariates rather than moderators. Although subgroup analyses for race and ethnicity are included, a more granular exploration of how specific socioeconomic and structural factors contribute to disparities could provide additional insights.”

Response:

We agree with the reviewer that moderator analysis offers further insight into disparities. While our initial analyses treated social determinants as covariates, we have added subgroup analyses stratified by sex, race/ethnicity, and age groups.

Comment: “Regarding the methods, please describe the inclusion and exclusion criteria for BRFSS participants.”

Response:

We have revised the Methodology section to clearly define the inclusion and exclusion criteria. Participants were included if they were aged 18 or older and had complete responses to the loneliness measure, all three outcome variables, and all covariates. Participants who responded “Don’t know,” “Refused,” or had missing data were excluded.

Comment: “Clarify how the survey design ensures national representativeness.”

Response:

We have added a paragraph explaining the BRFSS's complex multistage sampling design and the application of sampling weights to ensure state and national representativeness. These revisions are now detailed in the Methodology section.

Comment: “Further detail on how covariates were coded (e.g., categorical vs. continuous) would strengthen transparency.”

Response:

We have expanded the Covariates and Coding subsection to clearly indicate how each variable was treated in the analysis (e.g., age as continuous, education as categorical with dummy coding, etc.).

Comment: “Please provide stepwise equations for IPW estimation, including the propensity score model, how inverse probabilities were computed, and the final weighting approach.”

Response:

We have added a detailed, step-by-step explanation of the IPW process in the Statistical Analysis section, including the multinomial logistic regression used to generate propensity scores, computation of inverse probabilities, trimming of extreme weights at the 1st and 99th percentiles, and the final multiplication with BRFSS sampling weights.

Comment: “How were sampling weights from BRFSS incorporated into the IPW procedure? Were they applied multiplicatively? Did you implement strategies to address extreme propensity scores?”

Response:

Yes, the BRFSS sampling weight (pweight = _llcpwt) was applied multiplicatively with the IPW weights to create the final analysis weight. We have now clarified this process in the methodology and included code examples. We also describe our trimming of extreme weights at the 1st and 99th percentiles to reduce their influence.

Comment: “Additional details on how the ATE was computed and what the reported values represent in the context of loneliness and health outcomes would be helpful.”

Response:

We expanded the Interpretation of ATE Estimates subsection to explain how the ATE represents the average differences in outcomes between loneliness levels, adjusted for confounders. We provide clear examples (e.g., “10.8 more poor mental health days for ‘Always lonely’ vs. ‘Never lonely’ individuals”).

Comment: “Would you be open to providing the statistical code?”

Response:

Yes. We have stated in the revised manuscript that the full annotated Stata code is available upon request and will also be posted on a public GitHub repository upon acceptance for publication. This is now noted in the Software and Reproducibility section.

Comment: “The discussion effectively contextualizes the findings, but the contribution relative to prior studies could be made more explicit... Strengthening the discussion on mechanisms and policy implications would further enhance its contribution.”

Response:

We revised the Discussion to clearly articulate how our study builds on prior work—emphasizing the use of causal methods, nationally representative data, and the joint analysis of mental and physical health outcomes. We also elaborated on the biological and psychosocial mechanisms linking loneliness to health outcomes and expanded the section on policy and clinical implications to include practical interventions (e.g., social prescribing, digital tools, community investments).

Reviewer #2

Comment: “How was self-diagnosis avoided regarding depression? Was there a mechanism used to ensure that data collected regarding diagnosis is accurate?”

Response:

We appreciate this important question. The BRFSS depression item specifically asks whether a healthcare professional has ever told the respondent they had depression. This helps mitigate concerns about self-diagnosis and aligns the variable more closely with clinical depression. We have clarified this in the Discussion section under “Addressing Self-Reported Limitations.”

Once again, we thank both reviewers for their insightful comments, which have significantly improved the clarity and impact of the manuscript. We believe these revisions address all concerns and enhance the scientific rigor, transparency, and contribution of our work.

---

## [Decision Letter · Decision Letter 1]

The Impact of Loneliness on Depression, Mental Health, and Physical Well-being

PONE-D-25-04971R1

Dear Dr. Akinyemi,

We’re pleased to inform you that your manuscript has been judged scientifically suitable for publication and will be formally accepted for publication once it meets all outstanding technical requirements.

Kind regards,

Renata Komalasari

Academic Editor

PLOS ONE

Additional Editor Comments (optional):

The manuscript has addressed the reviewers' comments and my comments. It is appropriate for publication exactly as is.

Reviewers' comments:

Reviewer's Responses to Questions

**Comments to the Author**

1. If the authors have adequately addressed your comments raised in a previous round of review and you feel that this manuscript is now acceptable for publication, you may indicate that here to bypass the “Comments to the Author” section, enter your conflict of interest statement in the “Confidential to Editor” section, and submit your "Accept" recommendation.

Reviewer #1: All comments have been addressed

2. Is the manuscript technically sound, and do the data support the conclusions?

Reviewer #1: Yes

3. Has the statistical analysis been performed appropriately and rigorously? 

Reviewer #1: Yes

4. Have the authors made all data underlying the findings in their manuscript fully available?

Reviewer #1: Yes

5. Is the manuscript presented in an intelligible fashion and written in standard English?

Reviewer #1: Yes

6. Review Comments to the Author

Reviewer #1: The authors have addressed all of my concerns. I thank them for their responsiveness, especially providing the statistical code.

7. PLOS authors have the option to publish the peer review history of their article (what does this mean? ). If published, this will include your full peer review and any attached files.

**Do you want your identity to be public for this peer review?** For information about this choice, including consent withdrawal, please see our Privacy Policy .

Reviewer #1: **Yes: ** Tyler Reed Bell

---

## [Editor Report · Acceptance letter]

PONE-D-25-04971R1

PLOS ONE

Dear Dr. Akinyemi,

I'm pleased to inform you that your manuscript has been deemed suitable for publication in PLOS ONE. Congratulations! Your manuscript is now being handed over to our production team.

Kind regards,

on behalf of

Dr. Renata Komalasari

Academic Editor

PLOS ONE